materials science

ZnO:Sb, micro-wires, gas sensor, persistent photoconductance

**Author for correspondence:**
Khalid F. Eid
e-mail: keid@staff.birzeit.edu

This article has been edited by the Royal Society of Chemistry, including the commissioning, peer review process and editorial aspects up to the point of acceptance.

# Oxygen sensing with individual ZnO:Sb micro-wires: effects of temperature and light exposure on the sensitivity and stability

Tej Poudel Chhetri[1], Lei Kerr[2], Nada Masmali[1], Herbert Jaeger[1] and Khalid F. Eid[1,3]

[1]Department of Physics, and [2]Department of Chemical, Paper and Biomedical Engineering, Miami University, Oxford, OH 45056, USA
[3]Department of Physics, Birzeit University, Birzeit, Palestine

(iD) KFE, 0000-0002-7617-3464

Nanostructured ZnO has been widely investigated as a gas sensing material. Antimony is an important dopant for ZnO that catalyses its surface reactivity and thus strengthens its gas sensing capability. However, there are not enough studies on the gas sensing of antimony-doped ZnO single wires. We fabricated and characterized ZnO/ZnO:Sb core–shell micro-wires and demonstrated that individual wires are sensitive to oxygen gas flow. Temperature and light illumination strongly affect the oxygen gas sensitivity and stability of these individual wires. It was found that these micro- and nano-wire oxygen sensors at 200°C give the highest response to oxygen, yet a vanishingly small effect of light and temperature variations. The underlying physics and the interplay between these effects are discussed in terms of surface-adsorbed oxygen, oxygen vacancies and hydrogen doping.

## 1. Introduction

The wide-band-gap semiconductor ZnO is well suited for electrical resistance-based sensing applications, due mainly to its active surface, surface-dominated conductance and the relative ease of growing different ZnO structures [1–6]. This surface-sensitive behaviour has attracted an increased focus on ZnO nanostructures due to their large surface-to-volume ratios, which lead to a high tunability of their electric resistance. ZnO nano-wires [7,8] nanoparticles [9], nanorods [10,11] and thin films have been demonstrated to have potentially useful sensor

functions [12]. The cause of this change in resistance is still under debate and investigation. The effect is generally attributed to the presence of an oxygen vacancies-induced n-type conducting layer on the wire surface or to unintended hydrogen (H) substitution for oxygen. When a ZnO wire is exposed to oxygen, some oxygen atoms get adsorbed to the wire's conducting layer, thus either compensating the vacancies or interacting with H impurities and in either case significantly reducing the conductance of the wire [1,13–17]. Multiple sensing schemes and materials combinations have been proposed that revolve around the ZnO surface sensitivity to ambient gases [18–22]. Yet, light exposure and temperature increase also affect surface oxygen atoms and cause a large increase in conductance [23–25]. The ZnO surface eventually regains the oxygen after the removal of light illumination (or heat), thus resulting in the gradual decrease in conductance, known as persistent photoconductivity. These significant changes of ZnO surface conductance due to light illumination and temperature variations complicate the operation of a ZnO-based gas sensor [26]. Yet, ZnO gas sensors suffer some significant drawbacks, like the need to operate them at relatively high temperature (typically 350°C) and/or the need to use ultraviolet light to activate them at room temperature [27,28].

In addition, doping ZnO with certain elements might enhance its sensing properties such as selectivity and sensitivity [29,30]. Group-V elements, like antimony, have especially attracted research interest and debate, since they can be p-type dopants of ZnO [31–35]. Furthermore, some research suggests that Sb acts as a weak catalyst that increases the reactivity of the ZnO surface—thus enhancing its sensing ability [36–38]. While single nano- or micro-wires have promising sensing abilities, it was found that nano-wire–nano-wire junctions have superior sensitivity. The simplest way to fabricate these junctions is to grow core–shell nano-wires with a ZnO core and a metal-oxide shell [39,40]. The enhanced sensitivity in these core–shell wires is due to the formation of a charge depletion zone at the interface between the core and the shell. This depletion zone strongly influences the conductance of the wire and can be quite sensitive to doping and to adsorbed oxygen to the outer surface [41]. Yet, ZnO:Sb micro- and nano-wires have been poorly studied for gas sensing applications, especially the core–shell structures [23].

This article reports our systematic studies of the oxygen gas sensing ability of individual ZnO:Sb micro-wires and the optimal range of temperature and light illumination to improve their performance. More specifically, the report focuses on three main results. (i) An inexpensive approach to grow core–shell ZnO/ZnO:Sb micro-wires. (ii) The excellent sensing properties including improved sensitivity, lower temperature of operation and the absence of fluctuations in the resistance of the sensor as a result of changes in temperature or light exposure at a moderate temperature of 200°C. (iii) A systematic study of the effect of light exposure on sensing at different temperatures, which is lacking in the literature [23].

## 2. Experimental details

The ZnO:Sb micro- and nano-wires were grown by mixing powders of equal weights (e.g. 2 g) of SbO and Zn in a crucible, heating to 900°C for 2 h and then cooling down slowly to room temperature by turning the oven off. This produced ZnO:Sb micro-wires with diameters around 2 to 2.5 μm and lengths up to 3 cm. Their surface morphology was characterized using a ZEISS SUPRA 35 scanning electron microscope (SEM). The structural and phase properties of the wires were characterized by X-ray diffraction (XRD), energy-dispersive X-ray (EDX) spectroscopy and electron backscattered diffraction (EBSD). The latter two types of measurement were performed with the SEM as well. While individual wires were as long as 2–3 cm, we measured the resistance of sections under 1 mm in length to ensure the consistency of the experimental procedure. Individual ZnO:Sb wires were isolated, placed on a $SiO_2$-coated Si substrate and then connected in series to a D-cell battery and a 1.5 MΩ external resistor ($R_{Ext}$) via two gold wires of diameter 20 μm attached using H20E EPO-TEK silver conductive epoxy. The electrical resistance of the ZnO:Sb wire was then measured using a simple voltage divider by measuring the voltage drop across it using a National Instruments data acquisition unit (NI-USB 6211).

The temperature of the sample was measured with a thermocouple, and three LEDs of blue, red and green colours were used for light illumination. The intensity of the LED light was about 1.3 mW cm$^{-2}$ at a distance of 4 cm. The optical spectra of these LEDs were measured with an Ocean Optics USB4000 spectrometer. The LEDs were found to have wavelength spectra of 445–500 nm (blue LED), 485–560 nm (green LED) and 575–660 nm (red LED). The flow rate of oxygen was measured using a 565 Series 65 mm flowmeter.

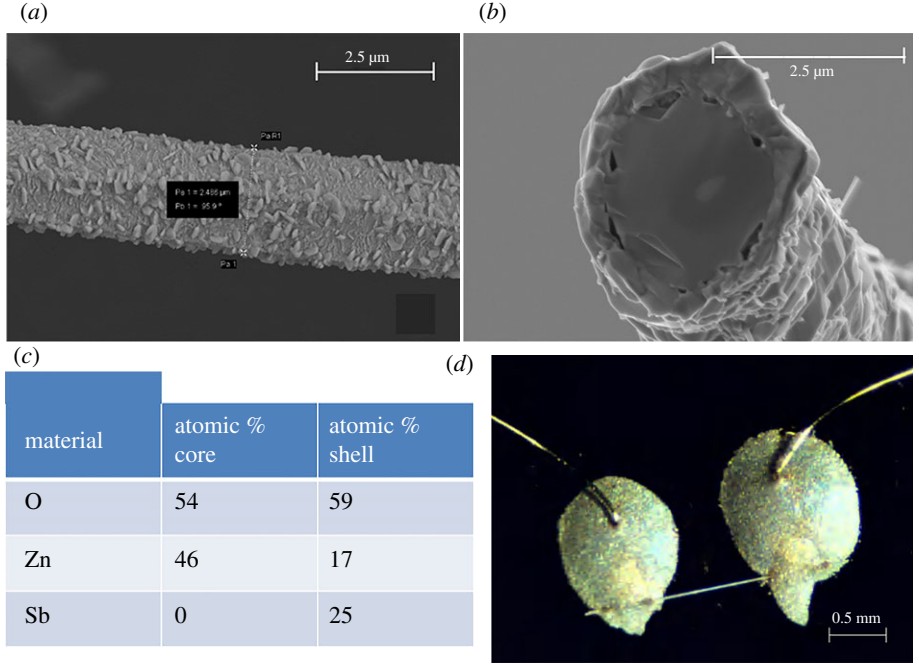

**Figure 1.** (*a*) SEM image for a selected individual ZnO:Sb wire. (*b*) Cross-sectional SEM image of a ZnO:Sb wire showing that it has a distinct core–shell structure. (*c*) Table showing the atomic composition of the ZnO core and the Sb-rich shell as determined from mapping EDX spectroscopy performed on the SEM. (*d*) An individual ZnO:Sb wire and two gold wires connected on a SiO$_2$ substrate with the help of conductive epoxy. The thickness of the gold wires is about 20 μm.

| material | atomic % core | atomic % shell |
|---|---|---|
| O | 54 | 59 |
| Zn | 46 | 17 |
| Sb | 0 | 25 |

## 3. Results and discussion

Figure 1*a* is an SEM micrograph of an individual ZnO:Sb wire of diameter approximately 2.5 μm. It is noticed that the micro-wire consists of grains with length varying from 200 nm to 600 nm. Our previous detailed studies of XRD, EDX and EBSD showed these wires to be polycrystalline, and to have certain areas mostly composed of ZnO, with others composed of Zn$_7$Sb$_2$O$_{12}$. Yet, the core–shell structure was not observed in these wires [42]. Another previous study found these ZnO:Sb wires to have n-type conduction [43]. Figure 1*b* is a representative cross-sectional SEM image of a micro-wire, which shows that the ZnO:Sb wires are composed of a core–shell heterojunction. This core–shell structure was observed in every wire we imaged with the SEM. Both EDX and EBSD studies of the core and the shell show that the core is basically undoped ZnO, while the shell is antimony-rich, as summarized in the table in figure 1*c*. The EDX studies showed clearly that the antimony-rich region (i.e. the outer shell of each wire) is composed of two compounds: ordonezite and zinc antimony oxide, which have a slightly larger band gap energy than ZnO [44,45], while we could not determine the ZnO core crystal structure using the same analysis [46]. EDX spectra observed in the core and those in the shell were both identical to those reported in [38]. Unlike other multi-step preparation methods, this surprisingly simple technique is capable of producing core–shell ZnO/ZnO:Sb micro-wires [47,48]. A peculiar observation in figure 1*b* is the several 'voids' between the ZnO core and the Sb-rich shell, measuring at tens of nanometres each. Such voids have been reported in ZnO nano-wires with Sb-based shells and are believed to be due to the presence of Zn$_7$Sb$_2$O$_{12}$ nano-scale precipitates in the shell. ZnO grows around these precipitates rather than directly in contact with them, thus creating the voids [49]. These Zn$_7$Sb$_2$O$_{12}$ areas lead to the stabilization of extra planes of oxygen atoms and have a significant impact on the charge depletion zone at the interface.

For photoconductance studies, we first connected a fresh ZnO:Sb wire on the SiO$_2$ substrate—as shown in figure 1*d*—to study the effect of light illumination. The wire was kept inside a small aluminium box ($2'' \times 1.5'' \times 1.5''$) placed directly on a hot plate, and a blue LED was fixed inside this box. The whole set-up was covered by a large cardboard box to block external light. After curing the conductive silver epoxy by annealing the sample at 150°C for 10 min, the sample was allowed to cool to room temperature and kept inside the cardboard box (i.e. in the dark) for 3 days to let the resistance of the wire attain its stable value. The blue LED inside the aluminium box was then turned on and the resistance of the wire was measured as a function of time, as shown in figure 2.

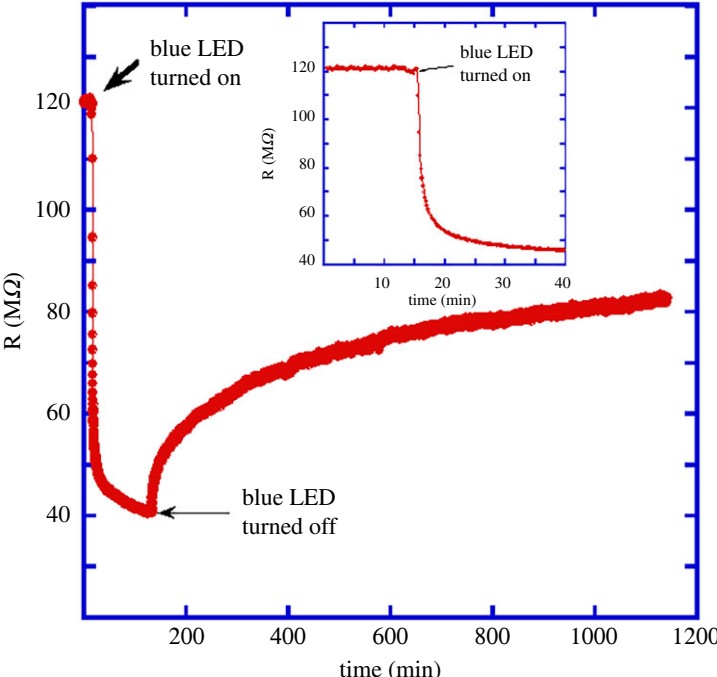

**Figure 2.** Photoconductivity of an individual ZnO:Sb wire under blue light illumination at room temperature. The inset shows the behaviour of the conductance during the first 40 min.

The resistance of the wire dropped rapidly when the light illumination started and reached a relatively stable value within 1 h. However, when the LED was turned off, the resistance increased very slowly following a short rapid increase and continued the slow rise for several hours. This slow increase cannot be attributed to the light-induced excitation of electrons from the valence band to the conduction band since the light energy is smaller than the energy band gap and the time scale of this rise takes up to tens of hours after the removal of the light illumination. Therefore, this slow increase of resistance after the LED was turned off is referred to as persistent photoconductance. As mentioned in the introduction, it is most likely related to the surface conductance that depends on the oxygen-vacancy and unintentional hydrogen doping concentrations at the outer shell of the wire. Absorbing light causes some oxygen atoms on and near the ZnO:Sb wire surface to get desorbed, thus leaving behind oxygen vacancies, which are known to be n-type dopants, as explained in the introduction. Turning the light off allows ambient oxygen atoms to slowly get adsorbed back onto the surface, thus filling oxygen vacancies, causing the surface doping density to decrease and leading to a slow increase in the wire resistance.

To investigate the effect of temperature on the resistance of these ZnO:Sb micro-wires, a fresh wire was connected following the same process discussed above and the sample was kept in darkness at room temperature in order to let the resistance stabilize. The temperature was then raised to 150°C and the resistance was recorded as shown in figure 3. The two important observations from this figure are the large drop in the resistance of the wire as the temperature increased, and that both the temperature and resistance took several minutes to reach their stable values.

A new ZnO:Sb wire was then connected following the same process as before and also left in darkness at room temperature until its resistance stabilized. The blue LED inside the aluminium box was then turned on and the resistance of the wire was measured at different temperatures from room temperature to 200°C. At each temperature, we waited for 90 min for temperature and resistance to stabilize and then measured the resistance. After finishing this set of measurements, the blue LED was turned off and the wire was kept in the dark for several days and then the resistance of the same wire was measured in the same manner by varying the temperature to obtain the temperature dependence of the resistance while the LED was off. Figure 4 shows only a slight decrease in the ZnO:Sb wire resistance when increasing the temperature under light illumination but a much stronger dependence when the light is off. $\Delta R/R$ is defined as the relative difference between the resistance measured without light illumination and that under blue LED light illumination. $\Delta R/R$ varies with temperature as shown in the inset. The most pronounced reduction in the resistance of the ZnO:Sb wire occurs when turning the light on at room temperature. However, at 200°C, the resistance was approximately equal whether the light was on or off as seen from figure 4. This result shows that the light

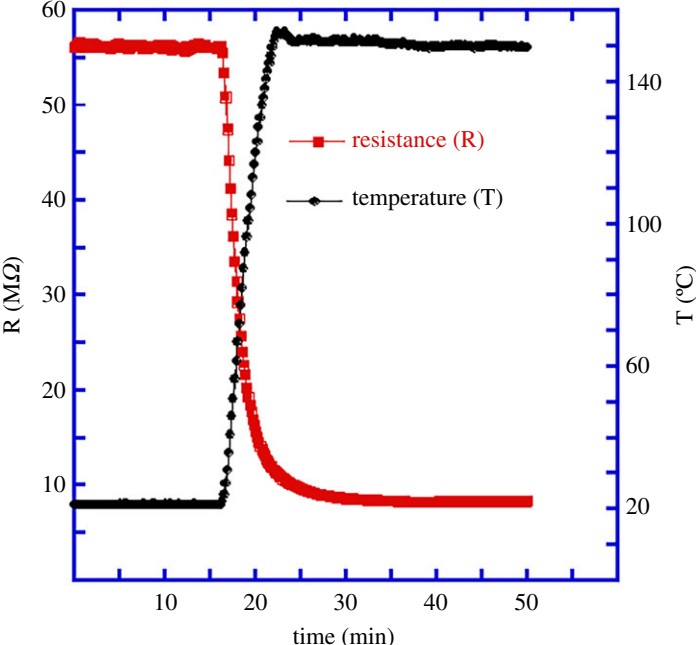

**Figure 3.** Resistance of an individual ZnO:Sb micro-wire as a function of temperature. Raising the wire temperature to 150°C causes its resistance to drop to less than 20% of its room temperature value.

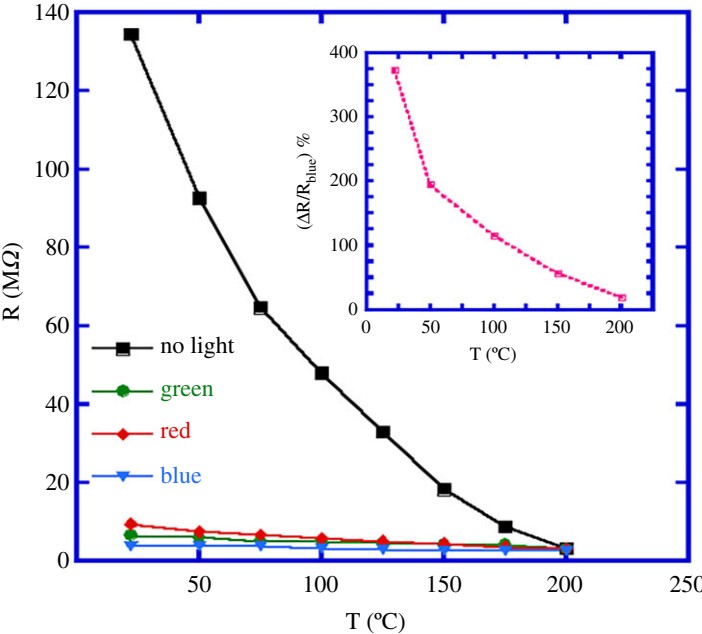

**Figure 4.** Temperature dependence of the wire resistance measured in the presence of blue LED light (blue curve), red LED light (red curve) and green LED light (green curve) and in the absence of any light (black curve). The inset is the temperature dependence of the relative change in the resistance with and without blue LED light.

illumination immensely affects the surface conductivity of ZnO:Sb wire at room temperature but has an insignificant effect at 200°C. It is also useful to notice that the drop in resistance associated with light exposure at room temperature is the same as that due to raising the temperature to 200°C without light. This suggests that the desorption of oxygen from the wire surface is responsible for the resistance variation in both cases. Under light illumination, the adsorbed oxygen is removed from the surface of the wire and a hydrogen-rich layer is left at the surface. The densities of these hydrogen dopants and possible oxygen vacancies are higher near the surface. These imperfections donate electrons which eventually increase the effective doping at the surface causing a drop in the resistance

(i.e. an increase in the conductance). All these phenomena on the surface of the ZnO:Sb wire are prominent upon light illumination at room temperature. However, there is no significant change of the conductance upon light illumination of the ZnO:Sb wire at 200°C. This is likely because the thermal energy already causes the oxygen atoms at or close to the surface to desorb, which is the same effect the light illumination causes at room temperature, so that there are no additional unreacted hydrogen atoms or oxygen vacancies to be created by light illumination at 200°C. Thus, we do not observe a significant effect on the resistance of the ZnO:Sb wire by light illumination at 200°C. The important aspect of this result is that ZnO:Sb wires can be used to make different kinds of chemical sensors with high reliability/stability to operate at 200°C or above. Since neither temperature nor light intensity fluctuations cause a significant change in the wire resistance, in the event of a resistance change due to a gas flow, one can be confident that the change is solely caused by the sensing materials.

Figure 4 also shows the temperature dependence of the resistance under different light colours. Three LEDs were placed near the ZnO:Sb wire, one at a time: blue, green and red, where each LED was kept 4 cm from the ZnO:Sb wire. The resistance of the freshly connected wire was first measured at different temperatures in the range from room temperature to 200°C in the absence of light. With this measurement done, the resistance of the wire was then measured again at all those temperatures under green light illumination and later red and then blue light illumination, with the wire left in the dark for 1 day between measurements with each LED. This experiment also showed the same effect of light illumination as before: there is a large reduction in the resistance of the ZnO:Sb wire at room temperature under light illumination, but the resistance is almost unchanged irrespective of light illumination of any colour at 200°C. Furthermore, the blue colour caused the largest reduction in the resistance at room temperature compared to green and red colours. It is not obvious whether this difference is due to the larger energy of the blue light or a slightly higher intensity/brightness of the blue LED.

Importantly, figure 4 also shows that the effects of light illumination and heating give the same change in resistance and likely have the same underlying physics: the desorption of oxygen atoms from the wire surface causes a change in its surface conductance. This surface-dominated conductance can also be altered by choosing the gas surrounding the wire; an oxidizing gas compensates for the surface oxygen vacancies and reacts with hydrogen impurities, thus leading to a rise in resistance, while a reducing gas has the opposite effect. This gas sensitivity allows for gas sensing applications. The results of the experiments done so far clearly suggest that such a ZnO:Sb gas sensor should be operated at 200°C, where neither light intensity nor temperature fluctuations affect the resistance of the wire. This is contrary to operating such a device at room temperature, where large changes in resistance are associated with temperature and light intensity fluctuations, thus overwhelming any potential sensing-related changes. This can be seen in figure 4, where the resistance in the absence of light (black squares) varies steeply with increasing temperature for low values of temperature. The curve then becomes almost flat at temperatures close to 200°C, where the rate of change in the resistance with temperature becomes much smaller. Furthermore, turning any of the LEDs on at 20°C causes a huge drop in resistance while the resistance is unchanged whether LED light is on or off at 200°C. So, the data shown as black squares overlap with the data for blue, red or green LEDs at 200°C. Yet, it is crucial that the gas sensitivity of such ZnO:Sb wires is not diminished at 200°C. To investigate the oxygen gas sensing properties of individual ZnO:Sb wires up to 200°C, a fresh ZnO:Sb wire was connected as before and kept in the dark for 1 day. The resistance was then monitored at 200°C while oxygen was passed into the aluminium box housing the wire at a rate of 1.54 litres per minute (LPM) through the inlet made on one side of the box. Oxygen flow continued for 2 min each time and the temperature was kept constant at 200°C. The results are shown in figure 5.

When an n-type material with an active surface and surface-dominated conductance is exposed to an oxidizing gas like oxygen, the resistance of the material increases, while the opposite behaviour is observed for p-type material. ZnO:Sb wires show n-type conduction, so their resistance increases when they are exposed to oxygen. The ZnO:Sb wire showed a strong, consistent response to oxygen flow; the resistance increased by 28% on average at 200°C as shown in figure 5a. The mechanism behind this significant increase in the resistance of the wire is again likely to be the adsorption of oxygen molecules on the surface of the wire when it is exposed to oxygen. This adsorbed oxygen causes the electron depletion region on the surface to grow by extracting electrons from the conduction band and trapping the conduction electrons [50]. Figure 5b shows the change in the resistance of the same wire due to oxygen gas flow under identical conditions at 200°C measured after 6 days. We found the resistance to increase by 29% on average, in close agreement with the first measurement 6 days earlier. This reproducibility in the resistance response to oxygen flow helps construct reliable ZnO:Sb micro-wire-based sensors.

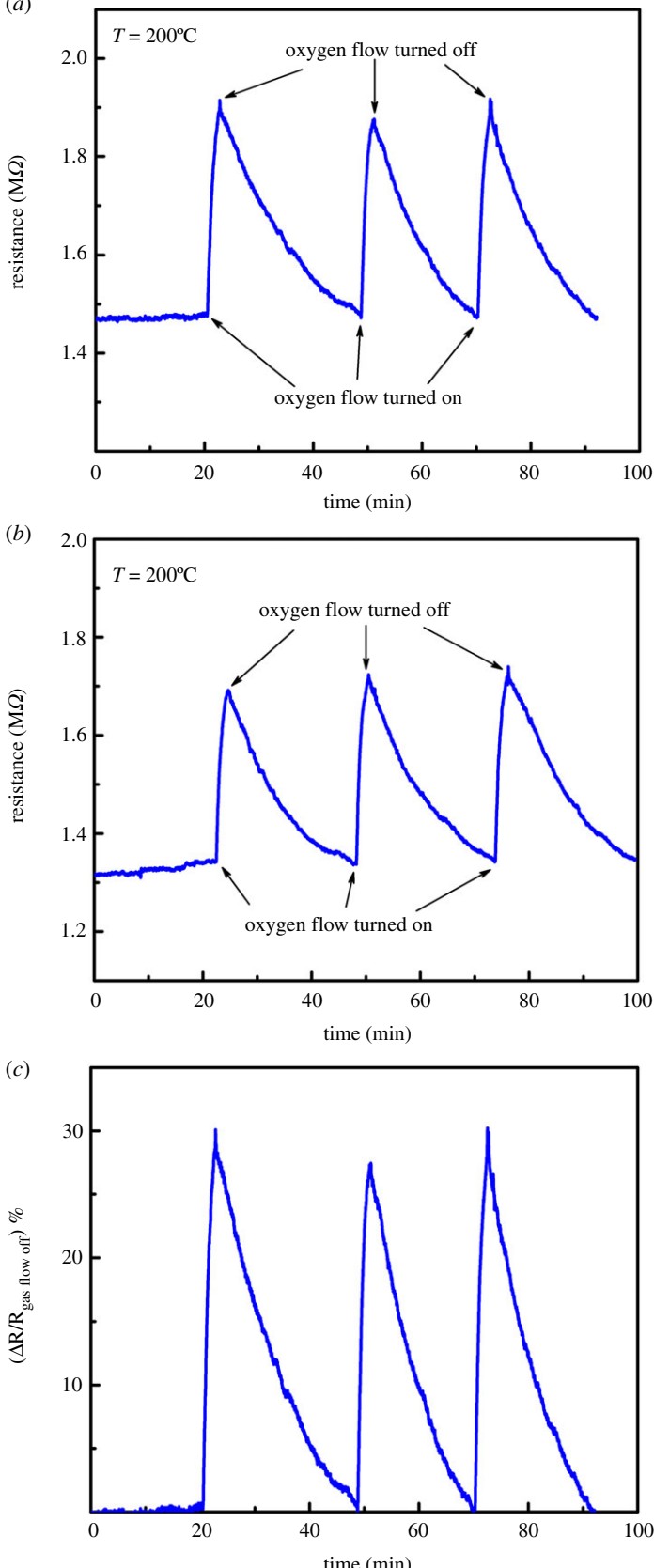

**Figure 5.** (*a*) Time variation of the resistance of an individual ZnO:Sb wire at 200℃ as oxygen gas flow is turned on and off repeatedly in the absence of light. (*b*) The resistance of the same wire measured after 6 days under the same conditions. The oxygen gas flow was on for 2 min each time at a rate of 1.54 LPM. (*c*) Time dependence of the relative change in resistance (ΔR/R) before and after oxygen gas flow calculated from (*b*).

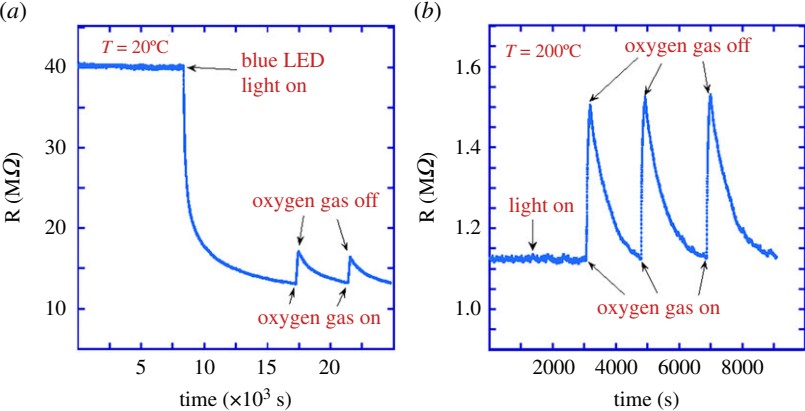

**Figure 6.** Resistance of ZnO:Sb wire to sense oxygen. (*a*) At room temperature, the effect of turning the light on or off is significantly larger than that of oxygen flow. (*b*) At *T* = 200°C, light has no measurable effect on the resistance while oxygen flow still causes a significant change, making oxygen sensing less susceptible to fluctuations and more reliable at 200°C.

Figure 6 shows the change in the electric resistance of a fresh ZnO:Sb wire as a function of time when oxygen gas flow is turned on and off as well as when light is turned on and off. It is apparent from the figure that light has a huge effect on the resistance at room temperature (figure 6*a*), the change of which is almost ten times larger than that due to oxygen flow. On the contrary, light has no effect at 200°C, while oxygen flow still causes a change in resistance of around 30%, as shown in figure 6*b*. Operating a ZnO:Sb-based resistive gas sensor at 200°C and above gives significantly more stable, reliable results than at room temperature. This is a significant improvement over other ZnO nano-wires that need to be heated to higher temperatures for optimal sensing performance and are more susceptible to erroneous sensing readings due to their sensitivity to light fluctuations [22,51]. Furthermore, this systematic study explains the role of light in resistive gas sensing at different temperatures, which provides vital information for applications [23,52]. Since ZnO is a wide-band-gap semiconductor, it is more capable of operating as a gas sensor at higher temperatures than those based on conventional semiconductors, such as Si [53].

## 4. Conclusion

In conclusion, we synthesized individual ZnO/ZnO:Sb core–shell micro-wires and demonstrated their use as reliable gas sensors with high sensitivity and stability at moderate temperatures (e.g. 200°C) but not at room temperature. Light illumination has a significant impact on the electrical conductance of the ZnO:Sb wires at room temperature but its effect decreases with an increase of temperature and becomes almost insignificant at 200°C or above. The oxygen-induced change in the resistance of ZnO: Sb wires was found to be stable.

Data accessibility. All data were collected using Labview software and saved as Excel (.xlxs) files. All four file names correspond directly to the figure numbers. Columns are labelled properly, where the first column is usually time. Other columns are either the resistance of the ZnO:Sb individual wire (in MΩ) or its temperature. The column is labelled properly in each case. Figure insets are meant to focus the attention on some specific parts of the data, but do not correspond to any 'new' datasets other than those used to produce the actual figures. Link to data: https://doi.org/10.5061/dryad.sj3tx965j.

Authors' contributions. All authors gave final approval for publication and agreed to be held accountable for the work performed therein.

Competing interests. We declare we have no competing interests.

Funding. No funding supported this research.

Acknowledgement. All SEM images were obtained and EDX and EBSD studies were conducted using Miami University's ZEISS SUPRA 35 SEM at the Center for Advanced Microscopy Imaging.

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
