## [Peer Review File · Royal Society Open Science]

Review History

RSOS-211243.R0 (Original submission)

Review form: Reviewer 1

Is the manuscript scientifically sound in its present form?

Yes

Are the interpretations and conclusions justified by the results?

No

Is the language acceptable?

Yes

Do you have any ethical concerns with this paper?

No

Have you any concerns about statistical analyses in this paper?

No

Recommendation?

Major revision is needed (please make suggestions in comments)

Comments to the Author(s)

In this manuscript, ZnO/ZnO:Sb core/shell micro-wires were synthesized and used for the sensing of oxygen. Some interesting results were obtained, but the conclusions were not fully supported by the data. Some drawbacks as listed below arose on reading the manuscript should be removed before its acceptance for publication.

1. The SEM images should be put in the section of results and discussion.
2. The formation of core/shell structure was not well characterized. The EDS elements mapping images can give direct evidence for this.
3. There were no characterizations to confirm the formation of Zn₇Sb₂O₁₂ in the shell. HRTEM and XRD can give some useful results.
4. Will the sunlight in the room affect the resistance of ZnO/ZnO:Sb core/shell micro-wires? Some experimental results should be given out.
5. The LED light illumination significantly reduced the resistance of ZnO/ZnO:Sb core/shell micro-wires. Authors should give some explanations to this phenomenon. The UV-vis absorption curves of ZnO/ZnO:Sb core/shell micro-wires should be given out.
6. Authors mentioned 'hydrogen doping concentrations at the outer shell of the wire' in the manuscript. However, the wire was not treated by hydrogen gas. So, where did the hydrogen come from?
7. Does the intensity of the LED light affect the resistance of the wire?
8. Why the resistance of the wire at 20 °C in Fig. 4 was different from that in Fig. 3?
9. In page 11, line 38 to 42, no data supported the claim 'where neither light intensity nor temperature fluctuations affect the resistance of the wire'.
10. As for Fig 6a, did the wire have response to the oxygen at 20 °C without light illumination?
11. The humidity can affect the response of sensing materials in practice. Therefore, the effects of humidity should be investigated.

Review form: Reviewer 2

Is the manuscript scientifically sound in its present form?

No

Are the interpretations and conclusions justified by the results?

No

Is the language acceptable?

Yes

Do you have any ethical concerns with this paper?

No

Have you any concerns about statistical analyses in this paper?

No

Recommendation?

Major revision is needed (please make suggestions in comments)

Comments to the Author(s)

Authors fabricated ZnO/ZnO:Sb core/shell micro-wires and used for the detection of oxygen. The studies show that these micro- and nano-wire oxygen sensors at 200°C give the highest response to oxygen. It is interesting. However, some issues need a major revision. It is not recommended in the present form.

1. In introduction part, recent references cited in this manuscript have not found. They are suggested to be added.
2. In experimental part, important information such as the amounts of raw materials, characterization instruments and their models. The part is suggested to rewrite in term of standardized writing format, and the results should be moved to III. RESULTS AND DISCUSSION part.
3. More characterizations and results are suggested to be added.
4. The quality of some Figures should be improved. For example, the font size in Fig.5
5. Some typos should be checked and revised. e.g. "there are little to no studies", (typically 350 C).
6. Property comparisons of ZnO/ZnO:Sb with reported materials are suggested to provide as a Table.

Review form: Reviewer 3

Is the manuscript scientifically sound in its present form?

Yes

Are the interpretations and conclusions justified by the results?

Yes

Is the language acceptable?

Yes

Do you have any ethical concerns with this paper?

No

Have you any concerns about statistical analyses in this paper?

No

Recommendation?

Accept with minor revision (please list in comments)

Comments to the Author(s)

This paper reports the fabrication of Sb-doped ZnO nanowires for oxygen sensing. The authors have investigated the effects of temperature and light exposure on the sensitivity and stability for oxygen sensing. The work provides new insights into the oxygen sensing properties of Sb-doped ZnO under the influence of temperature and light. However, some revisions are needed as detailed below:

1. The detailed mechanisms responsible for the decrease in resistance of Sb-doped ZnO nanowires can be provided.
2. Reg. the effect of light, does the pure ZnO nanowire sample also display similar decrease in resistance upon light exposure?
3. Stability data of the Sb-doped ZnO nanowires for oxygen sensing can be given.
4. The response time/recovery time of the oxygen sensing at 200 degrees C can be given/discussed.

5. The corresponding TEM images and TEM-EDS mapping of the Sb-doped ZnO nanowires can be provided. TEM can show the core-shell structure better than SEM and the TEM-EDS mapping can show the elemental distribution more clearly.
6. In the Introduction, some relevant references on the development of ZnO-based sensors for detecting different gases, such as *Materials and Design*, 191, 108628 (2020); *Sens. Actuators B* 261, 241-251 (2018); *Sens. Actuators B*, 191, 108628 (2020); *ACS Applied Nano Materials* 3 (9), 8982-8996 (2020); *Sens. Actuators B*, 209, 889-897 (2015) can be mentioned and cited to show the wide sensing applications of ZnO-based nanostructures.

Decision letter (RSOS-211243.R0)

Dear Dr Eid:

Title: Oxygen sensing with individual ZnO:Sb Micro-wires: Effects of temperature and light exposure on the sensitivity and stability
Manuscript ID: RSOS-211243

The editor assigned to your manuscript has now received comments from reviewers. We would like you to revise your paper in accordance with the referee and Subject Editor suggestions which can be found below (not including confidential reports to the Editor). Please note this decision does not guarantee eventual acceptance.

Please submit your revised paper before 22-Oct-2021. Please note that the revision deadline will expire at 00.00am on this date. If we do not hear from you within this time then it will be assumed that the paper has been withdrawn. In exceptional circumstances, extensions may be possible if agreed with the Editorial Office in advance. We do not allow multiple rounds of revision so we urge you to make every effort to fully address all of the comments at this stage. If deemed necessary by the Editors, your manuscript will be sent back to one or more of the original reviewers for assessment. If the original reviewers are not available we may invite new reviewers.

Yours sincerely,
Dr Ellis Wilde
Publishing Editor, Journals

RSC Associate Editor
Comments to the Author:
(There are no comments.)

RSC Subject Editor
Comments to the Author:
(There are no comments.)

Reviewers' Comments to Author:

Reviewer: 1

Comments to the Author(s)

In this manuscript, ZnO/ZnO:Sb core/shell micro-wires were synthesized and used for the sensing of oxygen. Some interesting results were obtained, but the conclusions were not fully supported by the data. Some drawbacks as listed below arose on reading the manuscript should be removed before its acceptance for publication.

1. The SEM images should be put in the section of results and discussion.
2. The formation of core/shell structure was not well characterized. The EDS elements mapping images can give direct evidence for this.
3. There were no characterizations to confirm the formation of Zn₇Sb₂O₁₂ in the shell. HRTEM and XRD can give some useful results.
4. Will the sunlight in the room affect the resistance of ZnO/ZnO:Sb core/shell micro-wires? Some experimental results should be given out.
5. The LED light illumination significantly reduced the resistance of ZnO/ZnO:Sb core/shell micro-wires. Authors should give some explanations to this phenomenon. The UV-vis absorption curves of ZnO/ZnO:Sb core/shell micro-wires should be given out.
6. Authors mentioned 'hydrogen doping concentrations at the outer shell of the wire' in the manuscript. However, the wire was not treated by hydrogen gas. So, where did the hydrogen come from?
7. Does the intensity of the LED light affect the resistance of the wire?
8. Why the resistance of the wire at 20 °C in Fig. 4 was different from that in Fig. 3?
9. In page 11, line 38 to 42, no data supported the claim 'where neither light intensity nor temperature fluctuations affect the resistance of the wire'.

10. As for Fig 6a, did the wire have response to the oxygen at 20 °C without light illumination?
11. The humidity can affect the response of sensing materials in practice. Therefore, the effects of humidity should be investigated.

Reviewer: 2

Comments to the Author(s)

Authors fabricated ZnO/ZnO:Sb core/shell micro-wires and used for the detection of oxygen. The studies show that these micro- and nano-wire oxygen sensors at 200°C give the highest response to oxygen. It is interesting. However, some issues need a major revision. It is not recommended in the present form.

1. In introduction part, recent references cited in this manuscript have not found. They are suggested to be added.
2. In experimental part, important information such as the amounts of raw materials, characterization instruments and their models. The part is suggested to rewrite in term of standardized writing format, and the results should be moved to III. RESULTS AND DISCUSSION part.
3. More characterizations and results are suggested to be added.
4. The quality of some Figures should be improved. For example, the font size in Fig.5
5. Some typos should be checked and revised. e.g. "there are little to no studies", (typically 350 C).
6. Property comparisons of ZnO/ZnO:Sb with reported materials are suggested to provide as a Table.

Reviewer: 3

Comments to the Author(s)

This paper reports the fabrication of Sb-doped ZnO nanowires for oxygen sensing. The authors have investigated the effects of temperature and light exposure on the sensitivity and stability for oxygen sensing. The work provides new insights into the oxygen sensing properties of Sb-doped ZnO under the influence of temperature and light. However, some revisions are needed as detailed below:

1. The detailed mechanisms responsible for the decrease in resistance of Sb-doped ZnO nanowires can be provided.
2. Reg. the effect of light, does the pure ZnO nanowire sample also display similar decrease in resistance upon light exposure?
3. Stability data of the Sb-doped ZnO nanowires for oxygen sensing can be given.
4. The response time/recovery time of the oxygen sensing at 200 degrees C can be given/discussed.
5. The corresponding TEM images and TEM-EDS mapping of the Sb-doped ZnO nanowires can be provided. TEM can show the core-shell structure better than SEM and the TEM-EDS mapping can show the elemental distribution more clearly.
6. In the Introduction, some relevant references on the development of ZnO-based sensors for detecting different gases, such as *Materials and Design*, 191, 108628 (2020); *Sens. Actuators B* 261, 241-251 (2018); *Sens. Actuators B*, 191, 108628 (2020); *ACS Applied Nano Materials* 3 (9), 8982-8996 (2020); *Sens. Actuators B*, 209, 889-897 (2015) can be mentioned and cited to show the wide sensing applications of ZnO-based nanostructures.

Author's Response to Decision Letter for (RSOS-211243.R0)

See Appendix A.

RSOS-211243.R1 (Revision)

Review form: Reviewer 1

Is the manuscript scientifically sound in its present form?

Yes

Are the interpretations and conclusions justified by the results?

Yes

Is the language acceptable?

Yes

Do you have any ethical concerns with this paper?

No

Have you any concerns about statistical analyses in this paper?

No

Recommendation?

Accept as is

Comments to the Author(s)

It is acceptable.

Review form: Reviewer 3

Is the manuscript scientifically sound in its present form?

Yes

Are the interpretations and conclusions justified by the results?

Yes

Is the language acceptable?

Yes

Do you have any ethical concerns with this paper?

No

Have you any concerns about statistical analyses in this paper?

No

Recommendation?

Accept as is

Comments to the Author(s)

The authors have addressed my previous comments thoroughly and conducted additional experiments necessary to improve the quality of this paper. Therefore, I am happy to accept the manuscript in the present form.

Decision letter (RSOS-211243.R1)

Dear Dr Eid:

Title: Oxygen sensing with individual ZnO:Sb Micro-wires: Effects of temperature and light exposure on the sensitivity and stability
Manuscript ID: RSOS-211243.R1

It is a pleasure to accept your manuscript in its current form for publication in Royal Society Open Science. The chemistry content of Royal Society Open Science is published in collaboration with the Royal Society of Chemistry.

Yours sincerely,
Dr Ellis Wilde
Publishing Editor, Journals

RSC Associate Editor
Comments to the Author:
(There are no comments.)

RSC Subject Editor
Comments to the Author:
(There are no comments.)

Reviewer(s)' Comments to Author:

Reviewer: 3

Comments to the Author(s)

The authors have addressed my previous comments thoroughly and conducted additional experiments necessary to improve the quality of this paper. Therefore, I am happy to accept the manuscript in the present form.

Reviewer: 1

Comments to the Author(s)

It is acceptable.

Appendix A

From: Khalid Eid, corresponding Author for manuscript: **RSOS-211243**

To: Dr. Ellis Wilde, Publishing Editor, Royal Society Open Science

Subject: Response Letter to Reviewers' comments on manuscript **RSOS-211243**

Dear Dr. Wilde,

Thank you and the reviewers of the Royal Society Open Science for the thorough and thoughtful review of our submitted manuscript entitled: **“Oxygen sensing with individual ZnO:Sb Micro-wires: Effects of temperature and light exposure on the sensitivity and stability”**.

We are pleased that all three reviewers found our work interesting and worthy of publication.

Please find attached our detailed response to every comment written by the reviewers. *We also added an ‘acknowledgment section’ at the end of the paper.*

I- REVIEWER 1

1- Reviewer 1 wrote: “The SEM images should be put in the section of results and discussion.”

Response: Figure 1 was moved to the ‘Results and Discussion’ section.

2- Reviewer 1 wrote: “The formation of core/shell structure was not well characterized. The EDS elements mapping images can give direct evidence for this.”

Response: We enhanced this discussion to emphasize the many SEM imaging, EDX, and EBSD studies we did and the similar results with those we already published in Ref. 38.

The text now reads: **“Our previous detailed studies of XRD, EDX, and EBSD showed these wires to be polycrystalline, and to have certain areas mostly composed of ZnO, while others composed of Zn₇Sb₂O₁₂. Yet, the core-shell structure was not observed in**

these wires.³⁸ Another previous study found these ZnO:Sb wires to have n-type conduction.³⁹ Fig. 1(b) is a representative cross-sectional SEM image of a microwire, which shows that the ZnO:Sb wires are composed of a core-shell heterojunction. This core-shell structure was observed in every wire we imaged with the SEM. Both EDX and EBSD studies of the core and the shell show that the core is basically undoped ZnO, while the shell is antimony-rich, as summarized in the table in Fig. 1(c). The EDX studies showed clearly that the antimony-rich region (i.e. the outer shell of each wire) is composed of two compounds: ordonezite and zinc antimony oxide, which have a slightly larger band gap energy than ZnO,^{40,41} while we could not determine the ZnO core crystal structure using the same analysis.⁴² EDX spectra observed in the core and those in the shell were both identical to those reported in Ref. 38.”

3- Reviewer 1 wrote: “There were no characterizations to confirm the formation of Zn₇Sb₂O₁₂ in the shell. HRTEM and XRD can give some useful results.”

Response: Our apologies for the lack of clarity here. We were trying to focus on the gas-sensing capabilities of these wires and ended up leaving some characterization details out. We did extensive EDX and EBSD analyses that were location-specific: either inside the core or in the shell area. These studies are what confirmed clearly the existence of Zn₇Sb₂O₁₂ in the shells. We modified the text to emphasize this- as seen in Response 2 above-, yet the spectra were again similar to what we reported in Ref. 38 earlier.

4- Reviewer 1 wrote: “Will the sunlight in the room affect the resistance of ZnO/ZnO:Sb core/shell micro-wires? Some experimental results should be given out.”

Response: Sunlight has a huge impact on these wires. It causes a larger drop in resistance than when exposed to the light of an LED due to the much larger intensity. Yet, we had much less control on sunlight intensity variations due to the time in the day and weather/visibility conditions. LEDs on the other hand offered us much better stability and control that were crucial in the study. This allowed us to focus on the effects of temperature, light color, and gas flow. Furthermore, sunlight is unlikely to be a factor in future sensing applications that usually include concealed sensors to exclude the effects of light fluctuations. We hope that the reviewer does not mind not including sunlight effects in the study.

5- **Reviewer 1 wrote:** “The LED light illumination significantly reduced the resistance of ZnO/ZnO:Sb core/shell micro-wires. Authors should give some explanations to this phenomenon. The UV-vis absorption curves of ZnO/ZnO:Sb core/shell micro-wires should be given out.”

Response: This behavior was ‘loosely explained’ in multiple places in the paper. We now included a detailed explanation in the first paragraph following Fig. 2. The last part of the paragraph now reads: “Therefore, this slow increase of resistance after the LED was turned off is referred to as the persistent photo-conductance. **As mentioned in the introduction, it is most likely related to the surface conductance that depends on the oxygen-vacancy and unintentional hydrogen doping concentrations at the outer shell of the wire. Absorbing light causes some oxygen atoms on and near the ZnO:Sb wire surface to get desorbed, thus leaving behind oxygen vacancies, which are known to be n-type dopants, as explained in the introduction. Turning the light off allows ambient oxygen atoms to slowly get adsorbed back onto the surface, thus filling oxygen vacancies, causing**

the surface doping density to decrease and leading to a slow increase in the wire resistance.”

6- **Reviewer 1 wrote:** “Authors mentioned ‘hydrogen doping concentrations at the outer shell of the wire’ in the manuscript. However, the wire was not treated by hydrogen gas. So, where did the hydrogen come from?”

Response: Thank you for the comment. As mentioned in the introduction, this topic is still under debate. The most ‘accepted’ explanation is the presence of oxygen vacancies near the surface. Yet, some theorists propose the possibility of “un-intentional hydrogen doping” due to a very weak contamination from the ambient air. This second explanation is less likely to be the cause of the photo-conductance, but we decided to include it since it is still a possibility and is still highly debated. We wrote in the introduction: “**The cause of this change in resistance is still under debate and investigation. The effect is generally attributed to the presence of an oxygen-vacancies-induced n-type conducting layer on the wire surface or to unintended hydrogen (H) substitution for oxygen. When a ZnO wire is exposed to oxygen, some oxygen atoms get adsorbed to the wire’s conducting layer, thus either compensating the vacancies or interacting with H impurities and in either case significantly reducing the conductance of the wire.**^{1, 13, 14, 15, 16, 17}”

7- **Reviewer 1 wrote:** “Does the intensity of the LED light affect the resistance of the wire?”

Response: Yes, the LED light intensity affects the wire resistance. For this reason, we always kept the LEDs at a fixed distance from the sample throughout this entire part of

our studies.

8- **Reviewer 1 wrote:** “Why the resistance of the wire at 20 °C in Fig. 4 was different from that in Fig. 3?”

Response: We studied many ZnO:Sb wires in this work. There is no way for us to control the exact length of a wire when using conductive epoxy to connect it to our measurement system. The irregularities on the surface –as shown in Fig. 1A)- also cause some variation. We felt it best for us as experimentalists to report the actual resistance (rather than a normalized quantity as, for example, R/R_0). The interested reader then has all the info to normalize the resistance, if desired. Where appropriate, we plotted $\Delta R/R$, as in the inset of Fig. 4 and in Fig. 5c).

9- **Reviewer 1 wrote:** “In page 11, line 38 to 42, no data supported the claim ‘where neither light intensity nor temperature fluctuations affect the resistance of the wire’.”

Response: This finding was not explained well. We now added the following detailed explanation immediately after the referenced lines. We wrote: “This can be seen in Fig. 4, where the resistance in the absence of light (black squares) varies steeply with increasing temperature for low values of temperature. The curve then becomes almost flat at temperatures close to 200°C , where the rate of change in the resistance with temperature becomes much smaller. Furthermore, turning any of the LEDs on at 20°C causes a huge drop in resistance while the resistance is unchanged whether LED light is on or off at 200°C. So, the data shown as black squares overlaps with the data for blue, red, or green LEDs at 200°C.” This effect becomes more obvious in Fig. 6b) and is the main reason for

our recommendation to operate these sensors at 200°C instead of room temperature.

10- **Reviewer 1 wrote:** “As for Fig 6a, did the wire have response to the oxygen at 20 °C without light illumination?”

Response: No, it did not have a significant response, only a weak response. The reason is that there is usually a lack of oxygen vacancies (abundance if adsorbed oxygen on the surface) at low temperature and without the light effect. So, adding more oxygen does not have any significant change in the resistance, since the surface is already saturated with oxygen. This effect is best explained in comment 5 above.

11- **Reviewer 1 wrote:** “The humidity can affect the response of sensing materials in practice. Therefore, the effects of humidity should be investigated.”

Response: Yes, the humidity can affect the results of this study. So, we kept the humidity fixed/controlled in our labs in order to isolate the effects of: temperature, light exposure, and oxygen gas flow.

II- REVIEWER 2

1- **Reviewer 2 wrote:** “In introduction part, recent references cited in this manuscript have not found. They are suggested to be added.”

Response: Thank you. We cited 5 recent papers (as also recommended by reviewer 3) to demonstrate the range of sensing applications of ZnO-based wires. We wrote: “**Multiple sensing schemes and materials combinations have been proposed that revolve around the ZnO surface sensitivity to ambient gases.**¹⁸⁻²²”

2- **Reviewer 2 wrote:** “In experimental part, important information such as the amounts of raw materials, characterization instruments and their models. The part is suggested to rewrite in term of standardized writing format, and the results should be moved to III. RESULTS AND DISCUSSION part.”

Response: Thank you. We moved the SEM images and characterization results to the ‘Results and Analysis’ section, leaving the general experimental details in the ‘Experimental Details’ section. We also included the specific amounts of the materials/reactants as well as the SEM tool model in this section. We wrote: “The ZnO:Sb micro- and nano-wires were grown by mixing powders of equal weights (e.g. 2 grams) of SbO oxide and Zn in a crucible, heating to 900°C for two hours and then cooling down slowly to room temperature by turning the oven off. This produced ZnO:Sb microwires with diameters around 2 to 2.5 μm and lengths up to 3cm. Their surface morphology was characterized using a ZEISS SUPRA 35 Scanning Electron Microscope (SEM),. The structural and phase properties of the wires were characterized by X-ray diffraction (XRD), Energy-dispersive X-ray (EDX) Spectroscopy, and Electron Backscattered Diffraction (EBSD). The latter two types of measurement were performed with the SEM microscope as well.” All other experimental details are included in the ‘Experimental Details’ section as well.

3- **Reviewer 2 wrote:** “More characterizations and results are suggested to be added.”

Response: We enhanced this discussion to emphasize the many SEM imaging, EDX, and EBSD studies we did and the similar results with those we already published in Ref. 38. The text now reads: “Our previous detailed studies of XRD, EDX, and EBSD showed these wires to be polycrystalline, and to have certain areas mostly composed of ZnO, while others composed of $\text{Zn}_7\text{Sb}_2\text{O}_{12}$. Yet, the core-shell structure was not observed in

these wires.³⁸ Another previous study found these ZnO:Sb wires to have n-type conduction.³⁹ Fig. 1(b) is a representative cross-sectional SEM image of a microwire, which shows that the ZnO:Sb wires are composed of a core-shell heterojunction. This core-shell structure was observed in every wire we imaged with the SEM. Both EDX and EBSD studies of the core and the shell show that the core is basically undoped ZnO, while the shell is antimony-rich, as summarized in the table in Fig. 1(c). The EDX studies showed clearly that the antimony-rich region (i.e. the outer shell of each wire) is composed of two compounds: ordonezite and zinc antimony oxide, which have a slightly larger band gap energy than ZnO,^{40,41} while we could not determine the ZnO core crystal structure using the same analysis.⁴² EDX spectra observed in the core and those in the shell were both identical to those reported in Ref. 38.”

4- **Reviewer 2 wrote:** “The quality of some Figures should be improved. For example, the font size in Fig.5”.

Response: We have improved the quality/clarity of Figures 5 as seen in the new draft.

5- **Reviewer 2 wrote:** “Some typos should be checked and revised. e.g. “there are little to no studies”, (typically 350 C).”

Response: We reviewed the entire paper and fixed both typos.

III- REVIEWER 3

1- **Reviewer 3 wrote:** “The detailed mechanisms responsible for the decrease in resistance of Sb-doped ZnO nanowires can be provided.”

Response: This behavior was ‘loosely explained’ in multiple places in the paper. We now included a detailed explanation in the first paragraph following Fig. 2. The last part of the paragraph now reads: “Therefore, this slow increase of resistance after the LED was turned off is referred to as the persistent photo-conductance. **As mentioned in the introduction, it is most likely related to the surface conductance that depends on the oxygen-vacancy and unintentional hydrogen doping concentrations at the outer shell of the wire. Absorbing light causes some oxygen atoms on and near the ZnO:Sb wire surface to get desorbed, thus leaving behind oxygen vacancies, which are known to be n-type dopants that increase the conductivity (decrease the resistance). Turning the light off allows ambient oxygen atoms to slowly get adsorbed back onto the surface, thus filling oxygen vacancies, causing the surface doping density to decrease and leading to a slow increase in the wire resistance.**”

2- Reviewer 3 wrote: “Reg. the effect of light, does the pure ZnO nanowire sample also display similar decrease in resistance upon light exposure?”

Response: Yes, pure ZnO wires show a similar effect. Most the literature available about persistent photo-conductance and the possible mechanism responsible for it has been obtained for nominally pure ZnO wires.

3- Reviewer 3 wrote: “Stability data of the Sb-doped ZnO nanowires for oxygen sensing can be given.”

Response: That is certainly an important point. Fig. 5 shows the change in the characteristics of one of the wires (response to gas and total resistance) over the period of 6 days. No

significant change was observed. Furthermore, some of these wires were tested several years after they were grown and we did not notice any changes.

4- Reviewer 3 wrote: “The response time/recovery time of the oxygen sensing at 200 degrees C can be given/discussed.”

Response: This is indeed a nice idea. Unfortunately, we did not carry a systematic study of the response/recovery times of these wires.

5- Reviewer 3 wrote: “The corresponding TEM images and TEM-EDS mapping of the Sb-doped ZnO nanowires can be provided. TEM can show the core-shell structure better than SEM and the TEM-EDS mapping can show the elemental distribution more clearly.”

Response: We did not have the capability of performing detailed TEM studies on these wires. Sample preparation was a significant challenge.

6- Reviewer 3 wrote: “In the Introduction, some relevant references on the development of ZnO-based sensors for detecting different gases, such as Materials and Design, 191, 108628 (2020); Sens. Actuators B 261, 241-251 (2018); Sens. Actuators B, 191, 108628 (2020); ACS Applied Nano Materials 3 (9), 8982-8996 (2020); Sens. Actuators B, 209, 889-897 (2015) can be mentioned and cited to show the wide sensing applications of ZnO-based nanostructures.”

Response: Thank you! We cited these 5 papers to demonstrate the range of sensing applications of ZnO-based wires. We wrote: “Multiple sensing schemes and materials combinations have been proposed that revolve around the ZnO surface sensitivity to ambient gases.¹⁸⁻²²”